# Tissue-Specific Metabolic Regulation of FOXO-Binding Protein: FOXO Does Not Act Alone

**DOI:** 10.3390/cells9030702

**Published:** 2020-03-13

**Authors:** Noriko Kodani, Jun Nakae

**Affiliations:** 1Division of Nephrology, Endocrinology and Metabolism, Department of Internal Medicine, Keio University School of Medicine, Tokyo 160-8582, Japan; nkodani@keio.jp; 2Department of Physiology, International University of Health and Welfare School of Medicine, Narita 286-8686, Japan

**Keywords:** FOXO, transcription factor, FOXO-binding protein

## Abstract

The transcription factor forkhead box (FOXO) controls important biological responses, including proliferation, apoptosis, differentiation, metabolism, and oxidative stress resistance. The transcriptional activity of FOXO is tightly regulated in a variety of cellular processes. FOXO can convert the external stimuli of insulin, growth factors, nutrients, cytokines, and oxidative stress into cell-specific biological responses by regulating the transcriptional activity of target genes. However, how a single transcription factor regulates a large set of target genes in various tissues in response to a variety of external stimuli remains to be clarified. Evidence indicates that FOXO-binding proteins synergistically function to achieve tightly controlled processes. Here, we review the elaborate mechanism of FOXO-binding proteins, focusing on adipogenesis, glucose homeostasis, and other metabolic regulations in order to deepen our understanding and to identify a novel therapeutic target for the prevention and treatment of metabolic disorders.

## 1. Introduction

Forkhead box (FOXO) transcription factors play important roles in apoptosis, the cell cycle, DNA damage repair, oxidative stress, cell differentiation, glucose metabolism, and other cellular functions [1]. FOXO genes were first identified in studies of chromosomal translocations in human tumors. FOXO1 was first identified in alveolar rhabdomyosarcoma [2], and FOXO3a and FOXO4 in acute leukemia [3,4,5,6]. FOXO6 was the last to be identified as possessing homology with other FOXO families [7]. The Foxo1-null mouse is embryonic lethal, whereas others are viable [8].

The transcriptional activity of FOXO is regulated by post-translational modification, including phosphorylation, acetylation, and ubiquitination, which determine subcellular localization, binding with DNA or other regulatory factors, and degradation, among other features. One of the basic regulatory systems is the growth factor-mediated phosphatidylinositol-3 kinase (PI3K)/Akt pathway, which induces phosphorylation of FOXO transcription factors. The three consensus AKT phosphorylation sites are conserved in all members of the mammalian FOXO families, except FOXO6 [7,9,10,11,12,13,14]. Phosphorylated FOXO proteins are negatively regulated by nuclear exclusion [10,11]. When Akt is in the inactivated state, FOXO proteins stay in the nucleus and regulate the expression of target genes.

FOXO proteins function primarily as transcription factors in the nucleus and bind to the FOXO binding consensus domain of target genes, regulating the expression of these genes. The highly conserved FOXO binding consensus domain, (G/C)(T/A)AA(C/T)AA, is identified as the FOXO-recognized element (FRE) [11,15]. Furthermore, FOXOs bind to partner proteins, including signaling molecules, transcription factors, and cofactors. The interactions between FOXO and its binding proteins demonstrate a synergistic effect to achieve complex and tightly regulated physiological activities that are specific to different tissues. Here, we focus on tissue-specific FOXO function in metabolism and describe the mechanism by which FOXO-binding proteins play a role in fine-tuning energy and glucose metabolism.

## 2. General Regulation of FOXOs by FOXO-Binding Proteins

### 2.1. 14-3-3 Proteins Bind to Phosphorylated FOXO and Suppress DNA Binding

14-3-3 proteins are a family of modulator proteins that regulate multiple signaling pathways by binding to Ser/Thr-phosphorylated motifs on target proteins [16,17,18]. 14-3-3 binds to FOXO3 as a dimer. In response to growth factors, the serine/threonine kinase AKT phosphorylates Thr32, Ser253, and Ser315 of FOXO3. Phosphorylation of Thr32 and Ser253 allows the binding of 14-3-3 dimer to FOXO3, resulting in masking the nuclear localization sequence (NLS) [19]. This was also shown in a crystallography structural study using a FOXO4 NLS model, in which 14-3-3 binding caused conformational changes on FOXO NLS, leading FOXO4 to translocate to the cytosol [20] (Figure 1). 14-3-3 binding to phosphorylated Thr32 and Ser253 sites also masks the DNA-binding domain of FOXO, leading to dissociation of FOXO from the FOXO-responsive element (FRE) [21,22,23,24].

In contrast, another protein kinase, the AMP-activated protein kinase (AMPK), is essential to the cellular responses to low energy level and activates FOXO function. AMPK has been shown to phosphorylate FOXO3 at six sites (Thr179, Ser399, Ser413, Ser555, Ser588, and Ser626) that are different from the AKT phosphorylation sites [25]. In the absence of growth factors and under inactivation of AKT signaling, FOXO3 is translocated into the nucleus. AMPK does not affect FOXO3 subcellular localization but phosphorylates the nuclear FOXO3 and activates transcription. In this pathway, FOXO3 is considered to sense the lack of growth factors and lack of energy, thus activating energy-producing pathways while inactivating the energy-consuming pathways [25].

Oxidative stress also plays an important role in the regulation of FOXO proteins. Under oxidative stress, FOXO translocates to the nucleus to gain stress resistance by activating genes involved in oxidative detoxification, such as manganese superoxide dismutase (MnSOD) and catalase [26,27]. The c-Jun N-terminal protein kinases (JNKs), a mitogen-activated protein kinase (MAPK) family that play a critical role in the regulation of stress, cell differentiation, and cell apoptosis, are involved in this process [28]. JNK phosphorylates FOXO4 at Thr447 and Thr451, inducing translocation of FOXO4 to the nucleus [29]. Another study demonstrated that JNK phosphorylates 14-3-3 proteins, which leads to dissociation of FOXO3 from 14-3-3 in the cytoplasm and induces FOXO3 translocation to the nucleus [30]. Another kinase, mammalian Ste20-like kinase-1 (MST1), phosphorylates FOXO3 at Ser212 [31] and FoxO1 at Ser207 [32], blocking the interaction of FOXO with 14-3-3 and inducing the nuclear localization of FOXO. MST1 has been shown to activate the JNK pathway in mammalian cells [33]. Under oxidative stress, the JNK and MST1 pathways activate FOXO and control whether the cells enter the survival pathway or apoptosis. 14-3-3 serves as a key protein in both the JNK/MST1 and AKT pathways (Figure 1).

### 2.2. Acetylation of FOXO by CREB-Binding Protein (CBP) is a “Hit and Run” Regulation

The cAMP response element-binding protein (CREB) binding protein (CBP) and its related protein p300 (p300/CBP) are histone acetyltransferases that act as coactivators of numerous transcription factors [34,35]. In the nucleus, FOXO1 binds to the target gene promoter, and CBP is recruited to form the CBP–Foxo1 complex. The CBP–Foxo1 complex activates transcription of target genes via nucleosomal histone acetylation and recruits a preinitiation complex containing RNA polymerase II to the gene promoter. Subsequently, CBP induces acetylation at the DNA-binding domain of Foxo1 and attenuates its DNA-binding ability [35,36]. This “hit and run” model of FOXO and CBP interaction enables an elaborately controlled expression of the target gene (Figure 2). When FOXO1 is phosphorylated at one of the three phosphorylation sites (N-terminal Akt motif), p300/CBP cannot bind FOXO1; instead, the domain becomes open to 14-3-3 binding, leading to nuclear exclusion [37].

### 2.3. Sirtuin Family Proteins Bind and Deacetylate FOXO, Leading to Nuclear Localization.

Silent information regulator 2 (Sir2) is an evolutionarily conserved histone deacetylase in the sirtuin family of nicotinamide adenine dinucleotide (NAD)-dependent deacetylases and is considered as a longevity regulatory gene. Life span extension by Sir2 was first reported in yeast [38]. In *Caenorhabditis elegans*, sir2 and FOXO ortholog daf-16 have overlapping and distinct functions in regulating life span [39,40]. Sir2 binds Foxo1 and deacetylates the CBP-acetylated residues (Lys242, Lys245, and Lys262) [35] and induces the transcriptional activity of Foxo1.

In mammals, the Sir2 family has seven members. Among them, SIRT1 is the closest homologue of yeast and *C. elegans* [41]. SIRT1 is a nutrient-sensing deacetylase activated by fasting and caloric restriction. SIRT1 is mainly localized in the nucleus, directly interacting with acetylated Foxo1 through the LXXLL motif (amino acids 459–463) [42,43], and deacetylation of Foxo1 transactivates a series of target genes [44,45]. Acetylation of Foxo1 by CBP increases the Akt-mediated phosphorylation of Foxo1 at Ser253, leading to translocation from the nucleus to the cytoplasm [36]. A study in hepatocytes showed that the Foxo1–Sirt1 interaction overrides Akt-mediated phosphorylation and keeps Foxo1 in the nucleus [46]. Expression of Foxo1 target genes is then increased, leading to activation of gluconeogenesis and increased glucose release from hepatocytes [46].

A study using mice with constitutively acetylated Foxo1 (*Foxo1^KQ/KQ^*) and constitutively deacetylated Foxo1 (*Foxo1^KR/KR^*) clearly demonstrated the significance of FOXO acetylation in vivo [47]. *Foxo1^KQ/KQ^* is predominantly cytoplasmic with a loss of function phenotype, whereas *Foxo1^KR/KR^* is predominantly nuclear with a gain of function phenotype. *Foxo1^KQ/KQ^* is embryonic lethal due to cardiac and angiogenic defects. *Foxo1^KR/KR^* mice present with hyperglycemia and insulin resistance. Increased hepatic gluconeogenic gene expression and decreased glycemic gene expression result in excessive hepatic glucose output. Furthermore, *Foxo1^KR/KR^* mice have decreased free fatty acid (FFA) and triglyceride (TG) levels with a lower respiratory quotient, which is consistent with a state of preferential lipid usage. These findings suggest that, in response to fasting or caloric restriction, deacetylated Foxo1 promotes gluconeogenesis, and with prolonged fasting, the gluconeogenesis shifts to lipolysis (Figure 3). Thus, acetylation of FOXO1 can be a “failsafe mechanism” to prevent excessive FOXO1 activity.

SIRT2 suppresses adipocyte differentiation. *Sirt2* expression is more abundant in adipocytes compared to other sirtuins. Studies in adipocytes have shown that caloric restriction, nutrient deprivation, and cold exposure stimulate *Sirt2* expression [48]. Sirt2 mainly localized in the cytoplasm binds and deacetylates Foxo1 [49,50]. Deacetylation of Foxo1 by Sirt2 promotes binding to peroxisome proliferator activated receptor (PPAR)γ, one of the main transcription factors regulating adipogenesis, and subsequent repression of PPARγ transcriptional activity suppresses adipogenesis (Figure 3) [50,51,52].

SIRT3 was initially observed in mitochondria, and later in the nucleus [53,54]. SIRT3 expression is activated by calorie restriction, and the increased expression of deacetylase in adipocytes induces the expression of genes involved in mitochondrial biogenesis [55]. Expression of SIRT3 has also been implicated in the synthesis and maintenance of cellular ATP levels in many tissues, including heart, liver, and kidney [56]. Sirt3 functions as a stress-responsive deacetylase that blocks the cardiac hypertrophic response by binding and deacetylating Foxo3 [57]. Deacetylated Foxo3 translocates to the nucleus, and the transcription of FOXO-dependent antioxidant genes, MnSOD, and catalase is activated (Figure 3). Reactive oxygen species (ROS)-mediated Ras activation is suppressed by MnSOD and catalase, and the subsequent downstream MAPK/ERK and PI3K/Akt signaling pathways are suppressed. This process contributes to suppressing the cardiac hypertrophic response.

### 2.4. Degradation of FOXO1 by Ubiquitination is Controlled by SKP2 Binding

Degradation of FOXO is regulated by ubiquitination. The PI3K/Akt pathway inhibits FOXO activity by promoting phosphorylation, followed by nuclear exclusion and subsequent proteasome degradation (Figure 4) [58]. Ubiquitin is transferred and covalently attached to FOXO1 via sequential activation of three enzymes, including ubiquitin-activating enzyme (E1), ubiquitin-conjugating enzyme (UBC E2), and ubiquitin ligase (E3). The SKP1–CUL1–F-box protein (SCF) complex is a multi-subunit RING-finger E3 ligase that targets FOXO1. CUL1 recruits adaptor protein SKP1 and the RING-finger protein RBX1, and functions as a scaffold protein. SKP1 binds to the F-box domain of F-box-containing proteins. SKP2 is one of the F-box-containing proteins and binds to FOXO1 in number of cell types. SKP2 binding to FOXO1 requires Akt-mediated phosphorylation of FOXO1 at Ser256, and SKP2 induces the polyubiquitination and degradation of FOXO1 [58,59].

## 3. Tissue-Specific Function of FOXO1-Binding Protein in Insulin Responsive Tissues

Tissue-specific regulation by FOXOs is achieved not only by tissue-specific expression of FOXOs but also by the fine-tuning of FOXO activity by FOXO-binding proteins. FOXO-binding proteins involved in metabolism in adipocytes, liver, pancreas, skeletal muscle, cardiac muscle and hypothalamus will be reviewed here. Redundancy and compensability exist among FOXO proteins. However, in most studies with tissue-specific Foxo-knockout mice, specific phenotypes are observed mostly in Foxo1-knockout mice, with the exception of CD4^+^ T cells [60]. Foxo1 functions involved in regulation of metabolism and energy expenditure have been well studied in mouse models [61]. This indicates that Foxo1 has evolved to possess distinctive functions in metabolism; therefore, Foxo1 and its binding partners will be focused on in this section.

### 3.1. Adipocyte

#### 3.1.1. FOXO1 Binding to PPARγ Antagonizes Its Function in Adipocytes

PPARγ is a ligand-activated nuclear receptor transcription factor that plays a pivotal role in the regulation of metabolism and inflammation [62]. PPARγ is expressed mainly in insulin-responsive tissues, where it plays a pivotal role in adipocyte differentiation and expression of adipose-specific genes [63,64]. FOXO1 functions as a transcription repressor by binding to the *PPAR*γ promoter [65].

FOXO1 also interacts with PPARγ. PPAR forms a heterodimer with the retinoid X receptor (RXR) to bind to the target region of DNA, and FOXO1 binding to PPARγ is considered to disrupt this PPARγ/RXR complex, resulting in an incapability of PPARγ to bind DNA [51]. PPARγ, on the other hand, antagonizes FOXO1 signaling, suggesting a reciprocal antagonistic interaction between FOXO1 and PPARγ [51]. In adipocytes, FOXO1 is anti-adipogenic, whereas insulin and PPARγ functions are pro-adipogenic [51,66]. FOXO1 activation in preadipocytes inhibits adipocyte differentiation, whereas PPAR functions in the opposite manner.

#### 3.1.2. Zfp238 Regulates the Thermogenic Program in Cooperation with Foxo1

Foxo1 is involved in energy homeostasis in adipose tissue [66,67,68,69]. Brown adipocytes are densely packed with mitochondria containing uncoupling protein 1 (UCP1) and are essential in thermoregulation. White adipocytes contain few mitochondria devoid of UCP1 and are involved in the storage and release of energy. Beige adipocytes contain densely packed mitochondria with UCP1 and are involved in thermogenesis. Beige adipocytes develop in white adipose tissue (WAT) in response to chronic cold exposure, exercise, β3-agonists, and PPARγ activity [70,71].

Zfp238 is a zinc finger-type transcription factor expressed in adipocytes. A recent study has shown that Zfp238 binds to Foxo1 and inhibits the transcriptional activity of Foxo1 in adipocytes [70]. The absence of Zfp238 in adipocytes results in the development of obesity, insulin resistance, and decreased energy expenditure. Foxo1 binds to the Foxo-binding element in the *Ucp1* enhancer region and inhibits *Ucp1* expression in adipocytes, suppressing energy expenditure. Under cold exposure and other stimuli, Zfp238 binds to Foxo1 and acts as a corepressor, leading to reactivation of *Ucp1* expression and the generation of beige adipocytes in WAT (Figure 5). These findings suggest that Zfp238 in adipose tissue regulates the thermogenic program in cooperation with Foxo1.

### 3.2. Liver

#### 3.2.1. PPARα Binding to FOXO1 Suppresses apoC-III Expression in the Liver

Hypertriglyceridemia is one of the main causes of atherosclerosis, leading to coronary artery disease and other vascular diseases. In hypertriglyceridemia, increased production of very low density lipoprotein (VLDL) and/or decreased clearance of TG-rich particles is observed. Adipolipoprotein C-III (apoC-III) inhibits lipoprotein lipase and hepatic lipase, impairing the clearance of TG-rich particles and leading to hypertriglyceridemia [72]. The *apoC-III* gene is one of the FOXO target genes. FOXO1 stimulates hepatic apoC-III expression, which is counteracted by insulin [73]. In insulin-resistant states, hepatic FOXO1 activity is increased, inducing transactivation of *apoC-III* and elevating plasma VLDL-TG levels. In a study of a high-fructose fed hamster, Foxo1 localized in the nucleus of hepatocytes induced overexpression of *apoC-III* and hypertriglyceridemia [74]. PPARα was shown to bind to and antagonize FOXO1 in hepatic *apoC-III* expression, suggesting the importance of FOXO1 deregulation in the pathology of insulin resistance and hypertriglyceridemia. These findings suggest that PPARα is required to prevent insulin resistance and hyperglyceridemia by suppressing FOXO1 activity.

#### 3.2.2. FOXO Binding to Hepatocyte Nuclear Factor-4 (HNF-4) Represses Expression of HNF-4 Target Genes

Hepatocyte nuclear factor (HNF)-4, a member of the steroid/thyroid nuclear receptor superfamily, is a transcription factor expressed in liver, kidney, intestine, and pancreatic β-cells [75,76]. HNF-4 binds to a specific DNA element as a homodimer and regulates expression of genes involved in glucose, cholesterol, and fatty acid metabolisms [77]. Foxo1 binding to HNF-4 represses HNF-4 transactivation [78]. Foxo1 interacts with the DNA-binding domain of HNF-4 and inhibits binding of HNF-4 to the DNA. The insulin signal pathway activates the PI3K/Akt signal and phosphorylates Foxo1, leading to decreased Foxo1-binding affinity to HNF-4, and HNF-4 transactivation is induced (Figure 6) [78]. This indicates that insulin regulates transcriptional activity of HNF-4 via Foxo1 as a transcriptional inhibitor.

Glucokinase (GCK) is one of the key enzymes that regulates glucose utilization in the liver. Several transcription factors are known to regulate *Gck* expression: PPARγ, hypoxia-inducible factor-1(HIF-1), sterol regulatory element-binding protein 1c (SREBP1c), and HNF-4 [79]. Acetylated Foxo1 does not bind to HNF-4, thus enabling HNF-4 binding to the *Gck* promoter, which induces *Gck* expression. Sirt1 deacetylates Foxo1 and deacetylated Foxo1 binds to the *Gck* promoter and disrupts the binding of HNF-4 to its binding site in the proximal *Gck* promoter [36]. These findings show that *Gck* expression in hepatocyte is tightly controlled not only by Foxo1–HNF-4 interaction but also by the post-translational modification of Foxo1 acetylation (Figure 6).

#### 3.2.3. PPARγ Coactivator 1a (PGC1α) Interaction with FOXO1 Is Essential in Insulin-Regulated Hepatic Gluconeogenesis

PGC1α is a transcription co-activator and regulates adaptive thermogenesis in brown fat, muscle-fiber specification, and hepatic-fasting response. In the liver, PGC1α induces gluconeogenesis during fasting [80]. When glucagon and glucocorticoids are elevated in the fasting state, expression of *Pgc1* is increased. PGC1α binds and activates FOXO1, leading to activation of gluconeogenic gene expression, such as glucose-6-phosphatase (*G6pc*) [81]. Foxo1 activation is suppressed by Akt-mediated phosphorylation, and Akt functions to specifically disrupt the FOXO1-PGC1α interaction. These findings indicate that the FOXO1 and PGC1α interaction is essential for executing insulin-regulated gluconeogenesis in the liver (Figure 7).

#### 3.2.4. FOXO1 Binding to C/EBPα Regulates Gluconeogenesis During Liver Development

CCAAT/enhancer-binding protein (C/EBP) is a family of basic-leucine zipper transcription factors involved in female fertility, gluconeogenesis, adipogenesis, and hematopoiesis [82]. Foxo1 directly interacts with C/EBPα and increases C/EBPα-dependent transcription activity [83]. During liver development, Foxo1 is up-regulated just before birth when gluconeogenic gene expression is induced. Foxo1 binds to the promoter of a target gluconeogenic gene, such as phosphoenolpyruvate carboxykinase (*Pck1*), and activates its expression in a C/EBPα-dependent manner. Insulin inhibits *Pck1* expression and other C/EBPα-dependent transcription enhanced by Foxo1. Thus, Foxo1 regulates gluconeogenesis by binding to C/EBPα, and by linking insulin signaling to C/EBPα during liver development [83].

#### 3.2.5. Sin3a Interaction with FOXO1 Confers Selective Regulation to Expression of G6pc and Gck in the Liver

FOXO1 inhibits glucose utilization by suppressing glucokinase (*Gck*) expression and increases glucose production by activating *G6pc* expression [84,85]. Although inhibition of FOXO seems beneficial for diabetes, FOXO inhibition will increase hepatic lipid synthesis and cause steatosis [86]. A recent study demonstrated how a single transcription factor controls both activation and suppression and regulates lipid and glucose homeostasis in hepatocytes (Figure 8) [87]. SIN3A is a pleiotropic corepressor involved in neoplastic transformation. The Sin3/HDAC corepressor complex functions in transcriptional regulation and is involved in major cellular events, such as cellular proliferation, differentiation, apoptosis, and cell cycle regulation [88]. SIN3A binds to the NH2-terminal half of FOXO1. FOXO1, SIN3a, and HDAC then form a complex and represses *Gck* expression in the liver. Ablation of SIN3A in the liver activates the expression of *Gck* without affecting other FOXO1 target genes, and reduces glycemia without concurrent steatosis. These findings suggest that FOXO1 selectively controls the expression of *Gck* by interacting with SIN3a. During insulin-induced *Gck* expression, clearance of SIN3a is observed, which requires phosphorylation of FOXO1 at two sites, Thr24 and Ser253. In contrast, inhibition of *G6pc* requires FOXO1 phosphorylation of one site, Ser253. These results are consistent with the previous finding that Thr24 phosphorylation is regulated by insulin and insulin-like growth factor-1 receptors [89], and that Thr24 and Ser253 kinases are distinct [90]. Identification of SIN3a as a FOXO1-binding protein confers the molecular mechanism of selective switching of FOXO1 target gene expression.

### 3.3. Pancreas

#### Foxo1 Corepressor (FCoR) Is a Main Regulator of Foxo1 Acetylation in Adipocytes and Pancreatic α- and β-cells

FCoR was identified as a Foxo1-binding protein in a yeast two-hybrid screen of a mouse 3T3-L1 cDNA library [70]. FCoR possesses a homologous sequence with the MYST family and exhibits acetyltransferase activity. FCoR interaction with Foxo1 suppresses Foxo1 transcriptional activity through Foxo1 acetylation and by preventing the Foxo1 interaction with deacetylase Sirt1. FCoR is phosphorylated at Thr93 by the catalytic subunit of protein kinase A and translocates to the nucleus. Therefore, FCoR can bind Foxo1 in both the cytosol and nucleus. FCoR acts as a "repressor" of Foxo1 and Foxo3a, but not of Foxo4 [70].

FCoR was first identified to possess an important role in adipocyte differentiation. In *FcorKO*, Foxo1 is activated and adipogenesis is inhibited. FCoR inhibits Foxo1, leading to increased *PPARg* expression, enhanced adipogenesis, and the generation of smaller adipocytes. In *FcorKO*, expression levels of inflammatory genes, including *Emr1* and *Ccr2*, are significantly increased in WAT, which explains the insulin resistance observed in *FcorKO*. Taken together, these data suggest that FCoR is a novel repressor that regulates insulin sensitivity and energy metabolism in adipose tissue by fine-tuning Foxo1 activity [70].

Furthermore, recent study showed that FCoR is also expressed in pancreatic α- and β-cells from the embryonic stage [91]. *FcorKO* mice exhibit glucose intolerance, decreased insulin secretion, and increased α-cell mass. Overexpression of β-cell-specific *Fcor* in the background of *FcorKO* rescued these phenotypes, suggesting that FCoR plays an important role in glucose homeostasis in β-cells. FCoR inhibits the expression of the *Aristaless-related homeobox* (*Arx*) gene, the α-cell determinant, through methylation of the *Arx* promoter region. In contrast, Foxo1 induces *Arx* expression by binding to the *Arx* promoter region. Foxo1 binding to the promoter region releases the DNA methyl transferase (DNMT)3a from the *Arx* promoter, and hypomethylation in the CG-rich promoter region is induced, leading to activation of *Arx* expression. In *FcorKO*, immunostaining shows an increase in Arx-positive endocrine cells from e15.5 and an increase in glucagon-positive cells from e17.5, suggesting that FCoR is involved in α- and β-cell differentiation from the endocrine progenitor stage. These findings suggest that *Arx* is a target gene of Foxo1 and its transactivation is suppressed by FCoR via methylation by DNMT3a.

FCoR acetylates Foxo1 and suppresses Foxo1 activity in islet cells [91]. The notable finding is that in *FcorKO*, most Foxo1 translocates to the nucleus and the intrinsic Foxo1 level decreases. This is probably due to deacetylated Foxo1 being activated in the absence of FCoR, followed by degradation. This result is consistent with the previous findings that activated FOXO is rapidly ubiquitinated and degraded [92]. Furthermore, in the absence of FCoR, lineage tracing analysis has shown β- to α-cell conversion [91]. This strongly suggests that FCoR is required to maintain β-cell identity, but further studies are necessary to identify the mechanism of function. These findings indicate that FCoR suppresses Foxo1 activity by acetylating pancreatic α- and β-cells, as well as adipocytes. This FCoR-Foxo1 axis regulates *Arx* expression and α-cell mass from the endocrine progenitor stage and is required to maintain α- and β-cell identity (Figure 9).

### 3.4. Smooth Muscle and Skeletal Muscle

#### 3.4.1. Foxo4 Interacts with Myocardin and Represses Smooth Muscle Cell Differentiation

Smooth muscle cells (SMCs) are unique cells with phenotypic plasticity and present transition between a quiescent contractile phenotype and a proliferative phenotype in response to local environmental cues, including growth factors/inhibitors, mechanical influences, cell–cell and cell–matrix interactions, and various inflammatory mediators [93]. Smooth muscle cells (SMCs) modulate their phenotype between proliferative and differentiated states in response to physiological and pathological cues. Myocardin is a transcriptional coactivator of smooth muscle genes. Foxo4 interacts with myocardin and represses SMC differentiation. Insulin-like growth factor-I stimulates differentiation of SMCs by activating PI3K/Akt signaling. Therefore, it is suggested that Foxo4 phosphorylation by PI3K/Akt signaling stimulates nuclear export of Foxo4, thereby releasing myocardin from its inhibitory effect, and promotes SMC differentiation [94]. 

#### 3.4.2. FOXO Binding to Csl in the Notch Pathway Controls Myogenic Differentiation and Fiber Type Specification in Skeletal Muscle

FOXO transcription regulates myogenesis not only in SMC, but also in skeletal muscle [95]. Conditional Foxo1 ablation in skeletal muscle resulted in increased formation of MyoD-containing (fast-twitch) muscle fibers and altered fiber type distribution at the expense of myogenin-containing (slow-twitch) fibers [96]. The Notch pathway plays an important role in neural, vascular, muscular, and endocrine differentiation during embryogenesis. Upon ligand-induced cleavage, the intracellular domain of the Notch receptor translocates to the nucleus, where it interacts with the DNA-binding protein CSL, a (CBF-1, Su(H), Lag-1)-type transcription factor, and changes its transcriptional properties from a suppressor to an activator of transcription [97]. Csl is a primary effector of the Notch pathway. Nuclear Foxo1 also binds to Csl and interacts with Notch. Binding of Foxo1 and Notch releases the binding of the two corepressors: nuclear corepressor (NCoR) and silencing mediator for retinoid and thyroid hormone receptor (SMRT), and recruits the coactivator mastermind-like 1 (Maml1), leading to activation of Notch target genes (Figure 10). Hairy and Enhancer of split (*Hes*) gene is a Csl target gene. *Hes* gene expression is activated by Notch/Foxo1 binding, and Hes1 function as a transcription factor that suppresses myogenic determination gene (MyoD) transcription [98]. Thus, Notch/Foxo1 binding to Csl may integrate the environmental cues through Notch and the metabolic cues through Foxo1, which regulate progenitor cell maintenance and differentiation.

### 3.5. Cardiac Muscle

#### Sirt1–Foxo1 Interaction Activates Autophagy Flux Under Energy Deficiency in Cardiac Muscle

Diabetic cardiomyopathy is defined by the existence of abnormal myocardial structure in the absence of other cardiac risk factors such as hypertension or coronary diseases. In cardiac muscle, insulin receptor substrates (IRS)-1 or IRS-2 activates the Akt signaling pathway and inactivates Foxo1, leading to promoting cardiac function and survival [99,100]. Under metabolic stress, such as hyperinsulinemia, p38α mitogen-activated protein kinase (MAPK) promotes degradation of IRS-1 and IRS-2 in cardiac myocytes and activates Foxo1, leading to cardiomyopathy and heart failure [99,101].

In contrast, Foxo1 plays an essential role in response to starvation. As mentioned in Section 2.3, under fasting or caloric restriction, Sirt1 deacetylates Foxo1 in hepatocytes and promotes gluconeogenesis. In cardiac metabolism, Sirt1 and Foxo1 interaction plays an essential role in starvation-induced autophagy [43]. Deacetylation of Foxo1 induces expression of genes involved in autophagy, including Rab7. Rab7 is a GTP-binding protein that enhances fusion of autophagosomes and lysosomes and stimulates autophagy flux. Autophagy flux activated by the Sirt1–Foxo1 pathway is beneficial for the heart during nutrient and energy deficiency [43]. Therefore, under metabolic stress, Foxo1 can cause diabetic cardiomyopathy, while it is essential to protect the heart under caloric restriction.

### 3.6. Hypothalamus

#### Sirt1 May Function as an Energy Sensor through FOXO1 Regulation in Hypothalamus

Foxo1 is a key regulator of insulin- and leptin-mediated food intake and energy expenditure in the hypothalamus. Previous studies have shown that anorexigenic proopiomelacortin (POMC) neurons and orexigenic agouti-related peptide (AgRP) neurons in the arcuate nucleus (ARC) of hypothalamus are the major neuropeptides involved in this process [102,103,104]. Foxo1 directly binds to the promoter of *Pomc* and *Agrp*. Leptin signaling activates Janus kinase 2 (JAK2) and the signal transducer and activator of transcription 3 (STAT3), leading to nuclear translocation of phosphorylated STAT3. STAT3 then activates the *Pomc* gene while it suppresses the *Agrp* gene, resulting in suppression of food intake (Figure 11). Insulin also suppresses food intake by activating the PI3K/Akt signal, inducing Foxo1 phosphorylation and translocation to the cytosol. This leads to activation of *Pomc* and suppression of the *Agrp* gene. FOXO1 and STAT3 function antagonistically by competing for the binding sites on the promoter region of *Pomc* and *Agrp* genes [103].

Sirt1 is expressed in POMC neuron and AgRP neurons in ARC [105]. For hypothalamic Sirt1 responses to food intake, however, the results are still controversial. Sirt1 expression level decreases with age and high-fat diet, and loss of Sirt 1 function in ARC causes dysregulation of energy balance [106,107]. Considering the interaction between Sirt1 and FOXO1 observed in other tissues, it is possible that Sirt1 and FOXO1 function to control food intake, however, further studies are necessary to clarify this mechanism [108].

## 4. Conclusions and Future Perspectives

The FOXO transcription factor is an evolutionarily conserved protein and presents a conserved mechanism of function from *C. elegans* to mammals. Post-translational modification of FOXO tightly controls the activation and/or repression of target gene expression in response to various external stimuli. In addition to the classical regulation by post-translational modifications, there are various regulatory systems involved in FOXO regulation. To achieve this elegant and complex regulatory mechanism, FOXO does not act alone. The partner proteins synergistically function to cope with versatile physiological changes in different organs. Previous studies have shown that there are a variety of patterns regarding interaction between FOXO and the FOXO-binding protein to regulate target gene transcriptions (Figure 12). The interaction with the tissue-specific FOXO-binding protein make possible the fine-tuning of FOXO activity.

The primary function of FOXO is to support the living organism to survive unfavorable conditions and attain longevity. However, in higher eukaryotes, we see that loss-of-insulin signaling, which causes gain-of-FOXO function, can lead to various metabolic disorders, including type 2 diabetes. This is the result of dysregulation of FOXO activity caused by conditions such as obesity and insulin resistance. FOXO function in “longevity” is featured in *C. elegans*, however, in mammalia, the fine-tuning of FOXO activity seems as if it is avoiding unnecessary longevity so as to maintain ecological integrity. Recent studies of single-nucleotide polymorphisms (SNP) analysis showed that FOXO3 is related to an extended life span in humans, while FOXO1 is not [109]. Loss of Foxo1 in mice is embryonically lethal, while loss of Foxo3 is not. These facts indicate that mammalia have evolved to gain diversified regulatory function of FOXO and that among other FOXOs, FOXO1 is the critical key for survival.

Understanding the multilayer mechanism of the tissue-specific metabolic regulation by FOXO and the fine-tuning mechanism by the FOXO-binding protein will lead us to make a proper approach to prevention and treatment of metabolic disorders.

## Figures and Tables

**Figure 1 cells-09-00702-f001:**
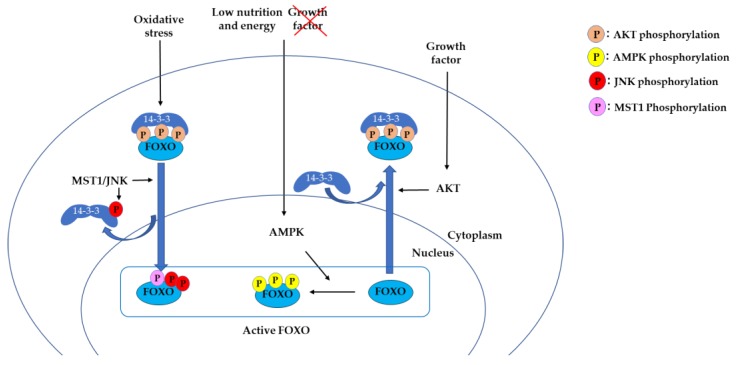
Transcriptional activity of forkhead box (FOXO) is regulated by the AKT, AMP-activated protein kinase (AMPK), and mammalian Ste20-like kinase/c-Jun N-terminal protein kinase (MST/JNK) pathways. In response to growth factor stimuli, AKT phosphorylates FOXO, leading to 14-3-3 binding and translocation of FOXO to the cytoplasm. In response to a low energy condition, AMPK phosphorylates FOXO in the nucleus and activates transcription. In response to oxidative stress, MST1/JNK phosphorylates FOXO, leading to nuclear translocation.

**Figure 2 cells-09-00702-f002:**
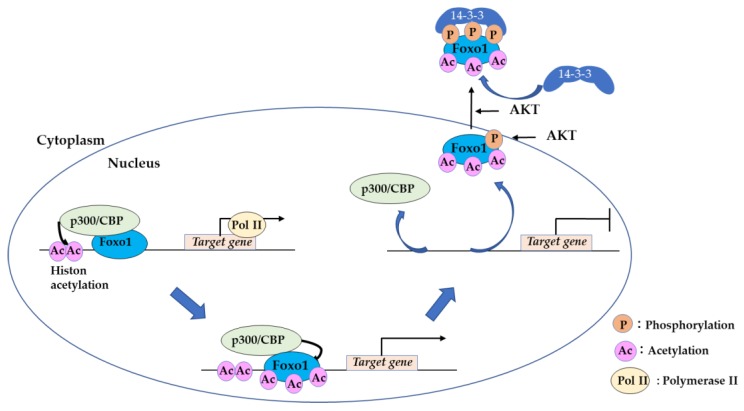
Transcriptional activity regulated by p300/CBP. The p300/CBP–Foxo1 complex activates target gene transcription by histone acetylation. Foxo1 is then acetylated by p300/CBP and becomes inactivated.

**Figure 3 cells-09-00702-f003:**
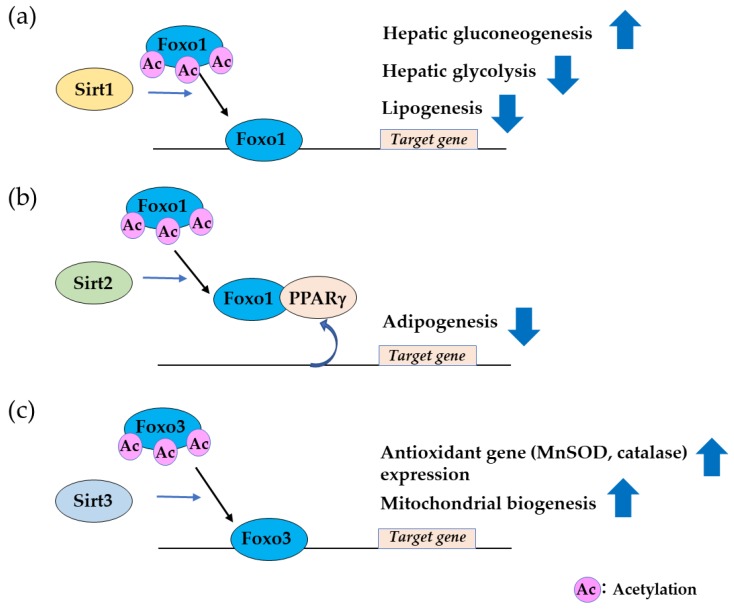
Sirtuin regulation of forkhead box (FOXO) in gluconeogenesis, adipogenesis, mitochondrial biogenesis and the antioxidant reaction. (**a**) Sirt1 binds and deacetylates Foxo1, leading to Foxo1 transactivation and induction of hepatic gluconeogenesis. (**b**) Sirt2 deacetylation of Foxo1 promotes binding to PPARγ, suppressing PPARγ activity and adipogenesis. (**c**) Sirt3 deacetylation of Foxo1 induces mitochondrial biogenesis and protects cells from oxidative stress.

**Figure 4 cells-09-00702-f004:**
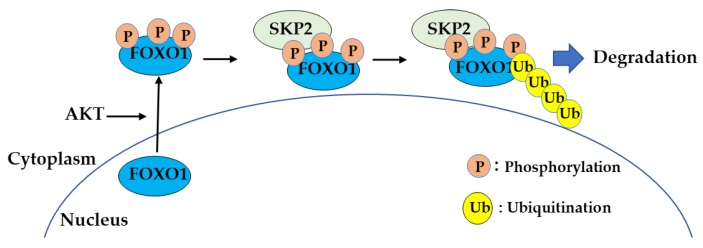
Forkhead box (FOXO) ubiquitination and degradation. Akt phosphorylation of FOXO and nuclear exclusion causes subsequent binding of SKP2, followed by ubiquitination and degradation.

**Figure 5 cells-09-00702-f005:**
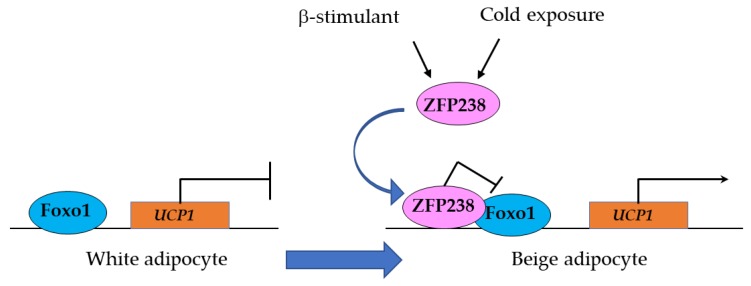
Zfp238 acts as a Foxo1 corepressor and regulates Ucp1 expression. In response to cold exposure and β-stimulant, Zfp238 binds to the promoter region of Ucp1 through Foxo1 and induces Ucp1 transcription.

**Figure 6 cells-09-00702-f006:**
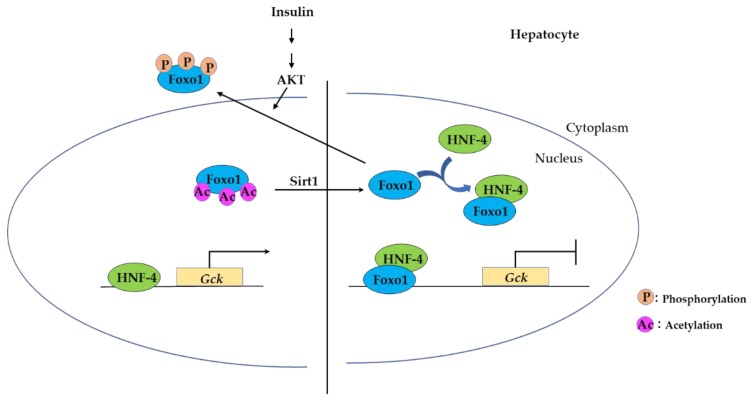
Foxo1 binding to hepatocyte nuclear factor (HNF-4) suppresses *Gck* expression. *Gck* expression in hepatocyte is tightly controlled not only by Foxo1–HNF-4 interaction but also by the post-translational modification of Foxo1 acetylation and phosphorylation.

**Figure 7 cells-09-00702-f007:**
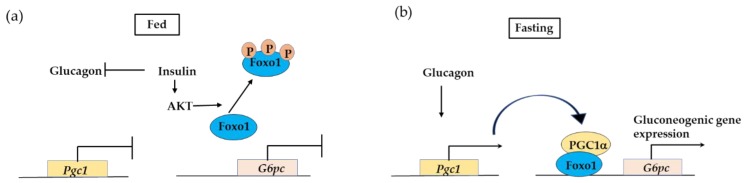
PGC1α–FOXO1 complex regulates expression of glucogenic genes. (**a**) Under the fed condition, insulin acts to suppress *Pgc1* expression and inactivate FOXO1 complex, leading to decreased expression of gluconeogenic genes. (**b**) Under the fasting condition, glucagon induces *Pgc1* expression leading to PGC1α–FOXO1 complex formation and induces glucogenic gene expression.

**Figure 8 cells-09-00702-f008:**
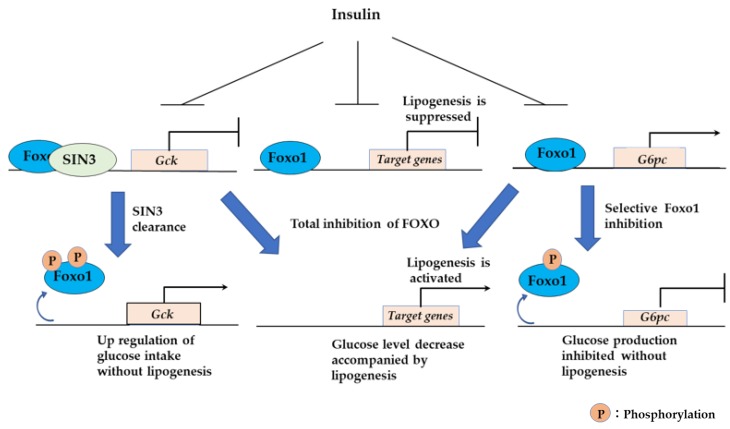
FOXO1 and SIN3 regulate the expression balance of *G6pc*, *Gck*, and lipogenesis genes. Insulin induces selective inhibition of FOXO1 and represses G6pc expression. SIN3 clearance selectively activates *Gck* expression; thus, hepatic glucose utilization is increased and the blood glucose level decreases. Selective inhibition of FOXO1 does not affect the expression of lipogenesis genes suppressed by FOXO1.

**Figure 9 cells-09-00702-f009:**
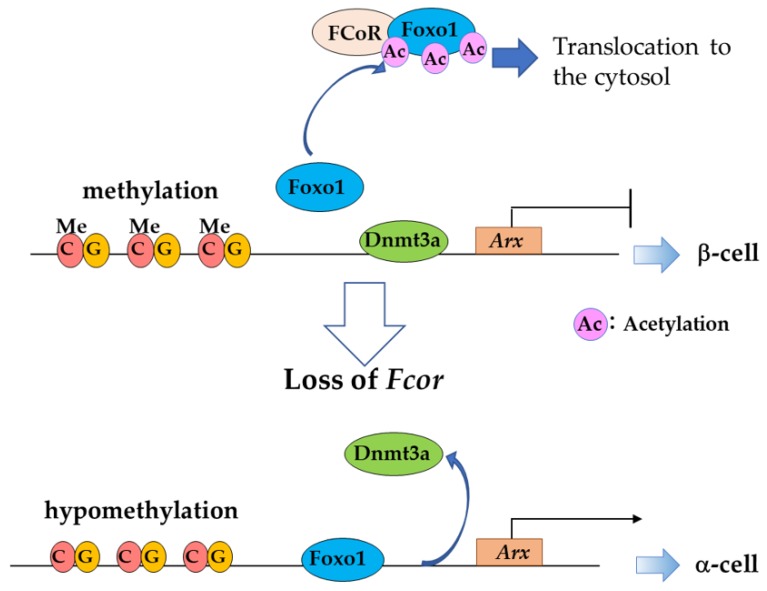
*Arx* expression is regulated by FCoR and Foxo1 interaction via methylation. FCoR suppresses Foxo1 binding to DNA and induces the binding of Dnmt3a to the promoter region of *Arx,* leading to methylation in the CG-rich region and suppression of *Arx* expression. In the absence of FcoR, Foxo1 binds to the promoter region and releases Dnmt3a, leading to hypomethylation of the CG-rich region and *Arx* expression induction.

**Figure 10 cells-09-00702-f010:**
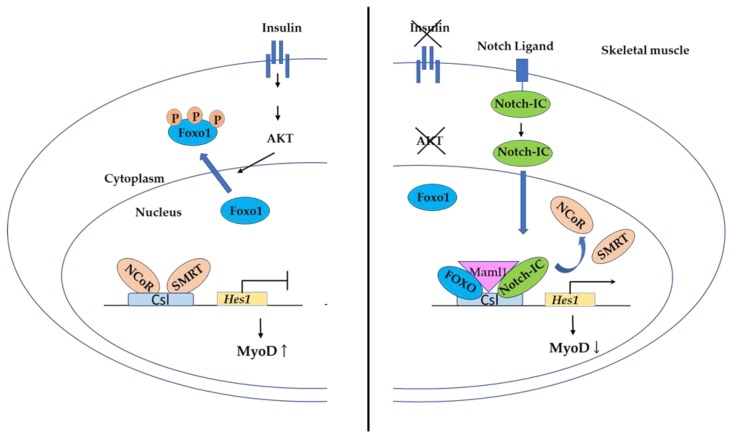
MyoD regulation by Notch/Foxo1 binding to Csl. Notch ligand binding to the Notch receptor releases the cellular domain of Notch (Notch-IC). The activated Notch-IC binds to the DNA-binding protein Csl. Activated Foxo1 also binds to Csl. Hairy and Enhancer of split (*Hes*) gene is a Csl target gene. Notch/Foxo1 activates *Hes1* transcription, leading to suppressing myogenic determination (*MyoD*) gene transcription. Nuclear corepressor (NCoR) and silencing mediator for retinoid and thyroid hormone receptor (SMRT) are transcription corepressors and mastermind-like 1 (Maml1) is a coactivator of *Hes1*.

**Figure 11 cells-09-00702-f011:**
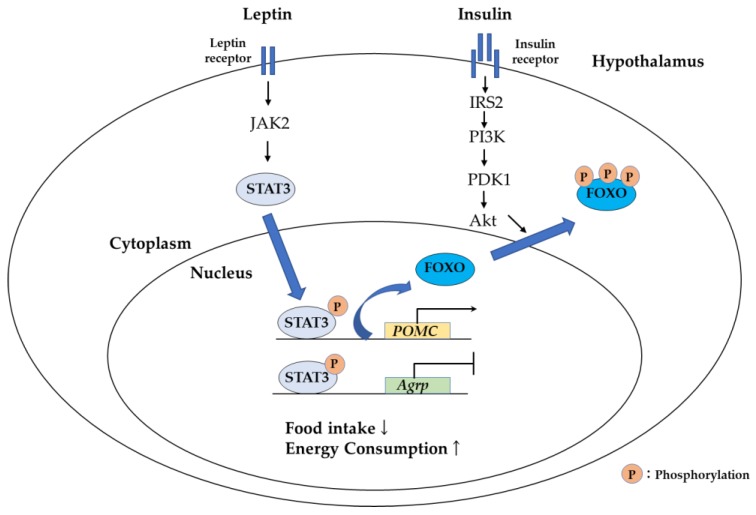
Food intake regulation by insulin and leptin in the hypothalamus. Insulin and leptin signaling decreases food intake and increases energy consumption. FOXO1 and STAT3 function antagonistically by competing for the binding sites on the promoter region of *Pomc* and *Agrp* genes. PDK1, 3-phosphoinositide-dependent protein kinase 1.

**Figure 12 cells-09-00702-f012:**
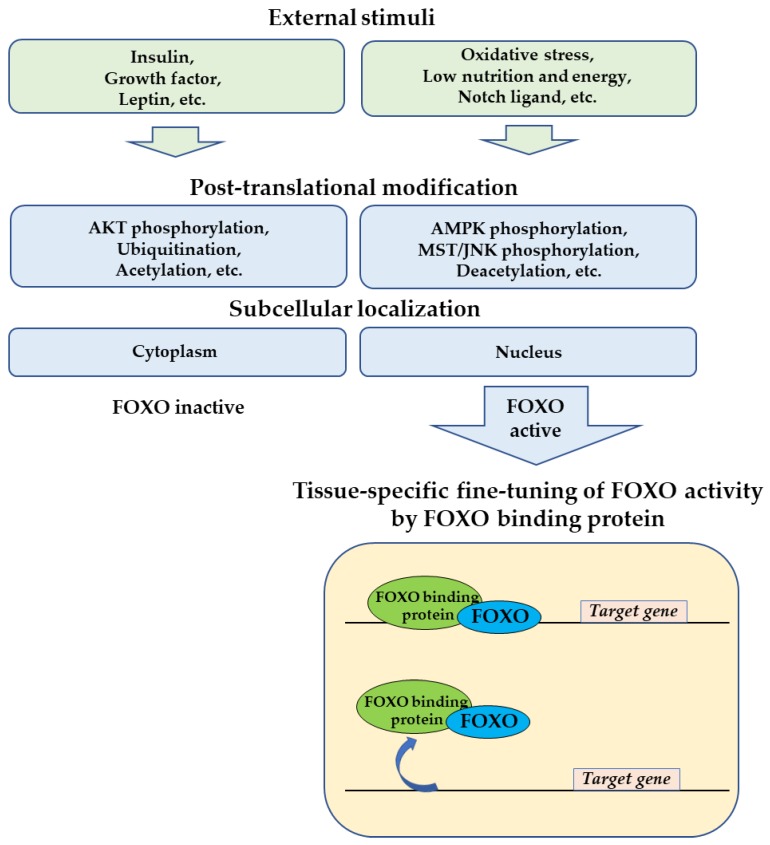
Multilayer regulation and fine-tuning of FOXO activity. In response to various external stimuli, post-translational modification of FOXO determines the subcellular localization and activity of FOXO. Tissue-specific transcription of the target genes by FOXO is fine-tuned by FOXO-binding protein.

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
