# Peer review of "Tissue-Specific Metabolic Regulation of FOXO-Binding Protein: FOXO Does Not Act Alone"

_cells, 2020, doi:10.3390/cells9030702_

Round 1

Reviewer 1 Report

This study reviewed the roles of FOXO binding proteins in a tissue-specific manner, suggesting the versatility of FOXO in the regulation of metabolic action in which largely rely on FOXO binding proteins.

The authors focused on adipocyte, liver, and pancreas to reveal the importance of FOXO-binding proteins that associated with adipogenesis, glucose homeostasis, and other metabolic regulations.

Overall, the review is well written with a logical progression of evidence, a combination of descriptive figures presented, and detailed legend is straightforward, therefore, I have a few minor comments below.

FOXO family has significant roles in other metabolic organs such as the hypothalamus, skeletal muscle, etc., it would be beneficial to broader audiences who are interested in the metabolic roles of FOXO and its binding proteins in these tissues as well. A review is not only a collection and conclusion from the literature. It requires in-depth thought from the authors for future research, guidance for other researchers, and an overview of the development of this field. The review has room to improve it. 

Author Response

Thank you very much for pointing out the important matter regarding this review.

We agree to include smooth muscle, skeletal muscle, cardiac muscle and hypothalamus in this review. Our first thought was that it would be better to focus on the main tissues, adipocyte, liver and pancreas, involved in metabolic regulation, however, function of FOXO in other tissues bear significant role as well. Therefore, we added “3.4 Smooth muscle and skeletal muscle”, “3.5. Cardiac muscle” and “3.6. Hypothalamus” in this review.

Regarding the significance of this review to the readers, we revised “4. Conclusion and future perspectives” so that this review is not a mere collection from the literature and can provide our thought regarding FOXO function. This manuscript was written in the purpose to comprehensively understand the function of FOXO from the point of FOXO binding proteins, since this enables a single transcription factor to play key roles in broad regulation of metabolism in different tissues. FOXO function is a double edge sword. While FOXO protects cells from outside stresses, it can cause unfavourable damage. These facts teach us the importance of fine tuning of FOXO activation making possible the regulation of transcription of numerous number of the target genes. To make an approach to prevention and treatment of metabolic disorders, understanding this complicated mechanism is essential, and thus, we made a review in this manuscript.

Reviewer 2 Report

The review by Kodani et al., has merit with respect to understanding the role of FOXO-binding proteins in cell physiology and metabolic disorders.

We suggest that the authors expand more about the role of FOXO in the brain heart, and in types of cancer (we see some references related to cancer, but the role of FOXO, in the context of cancer, is absent). The authors should be also aware that the topic has been addressed in other review articles and this is redundant to a certain extent (e.g., https://www.liebertpub.com/doi/full/10.1089/ars.2010.3370 has not been cited in this review)

Author Response

Thank you very much for your important comments regarding this review.

As pointed out, we have expanded the tissue in which FOXOs play an important role. Our first thought was that it would be better to focus on the main tissues, adipocyte, liver and pancreas, involved in metabolic regulation, however, functions of FOXOs in other tissues bear significant role as well, and we fully agree with your comment. Therefore, we added “3.4 Smooth muscle and skeletal muscle”, “3.5. Cardiac muscle” and “3.6. Hypothalamus” in this review.

Regarding cancer, we understand that FOXO transcription factors play key roles in cancer progression and metastasis. We believe that revealing the FOXO involvement in tumorigenesis and cancer progression is essential, especially in the development of antitumor agents and understanding the drug resistance and sensitivity among individuals and cancer type. However, we considered that it would be better not to include this broad topic in our review, in which the role of FOXO binding proteins involved in metabolic regulation is focused on. We considered that it would be better that this topic be discussed in other Reviews, in which FOXO function in cancer is the main theme.

The topics that we discuss maybe addressed in other review articles, however, we would like to discuss the function of FOXO form the aspect of FOXO binding proteins. We believe that the function of FOXO can be considered from a different point of view in this review.

We have cited “Targeting Forkhead BoxO1 from the Concept to Metabolic Diseases: Lessons from Mouse Models” (Antioxid Redox Signal. 2011; 14(4) 649-661) (No.56) in the beginning of Section “3. Tissue-specific function of FOXO1 binding protein in insulin responsive tissues”.   We consider that this is an important article in which studies of Foxo1 in mouse models are thoroughly discussed, and we should have cited this.

Reviewer 3 Report

As an informed, but non-expert researcher in FOXO, I was interested in reviewing this article as I felt it would reinforce my undestanding of the basic concepts of FOXO regulation and provide me with detail of how FOXO regulates different metabolic pathways.

However, I struggled with the basic concepts at the beginning of the review and could not build on this to grasp the more complex regulatort circuits described later in the article.

In particular;

Lines 54-55; the authors say that 14-3-3 binding to phospho-FOXO3 masks the DNA binding domain and this leads to translocation TO the nucleus and inactivation of FOXO3 function. Surely this is wrong and phospho-FOXO3 is retained in the cytoplasm by 14-4-3, as suggested in Figure 1.

Line 60-62: the authors state that AMPK does not affect subcellular localisation but "keeps FOXO3 in the nucleus"- surely keeping FOXO3 in the nucleus is the very definition of altering sub-cellular localisation!

These statements confused me very much and I could not then follow the arguments later in the review. The absence of simply explained concepts made the rest of the review feel like a list that I could not engage with.

These errors might be language problems or indeed my misunderstanding. However, failure to convey a key concept in simple terms defeats the purpose of a review.

Author Response

Thank you very much for your important comments regarding this review.

Line 54-55: This is an error and it should be “leading to translocation to the cytosol”. I truly apologize for the basic error and causing trouble understanding.

Line 60-62: I have rewritten the sentence so that it should be readable. “In the absence of growth factors and under inactivation of AKT signaling, FOXO3 is translocated into the nucleus. AMPK does not affect FOXO3 subcellular localization but phosphorylates the nuclear FOXO3 and activates transcription. ”  I have also revised Figure 1, to clearly show the mechanism by AMPK.

This manuscript was written in the purpose to comprehensively understand the function of FOXO from the point of FOXO binding proteins, since this enables a single transcription factor to play key roles in broad regulation of metabolism in different tissues. FOXO function is a double edge sword. While FOXO protects cells from outside stresses, it can cause unfavourable damage. These facts teach us the importance of fine tuning of the target genes. To make an approach to prevention and treatment of metabolic disorders, understanding this complicated mechanism is essential, and thus, we made a review in this manuscript. I apologize for the premature manuscript making difficult to understand the contents, and I hope this revised version can show you the concept of this review.

Round 2

Reviewer 3 Report

  1. it would be useful to clarify the role of 14-3-3 proteins in masking nuclear localisation signals and how nuclear transport is hence regulated.
  2. Figure 1- why not include activity of FOXO? e.g. which form of nuclear phospho-FOXO regulate gene expression? (or do all forms do this?).
  3. How good is the data with Resveratrol?- the life extending properties of this compound are frequently celled into question.
  4. Some formatting problems (paragraph width) around lines 409-411 and 455-463.
  5. C.elegans should be in italics (line 525).
  6. The statements about "ecological integrity" in the conclusion are beyond the scope of this type of review.

Author Response

Thank you for your comments. We have made revisions as follows;

1. It would be useful to clarify the role of 14-3-3 proteins in masking nuclear localisation signals and how nuclear transport is hence regulated.

We agree with this suggestion and have rewritten the 1st paragraph of section 2.1. How 14-3-3 binding masks NLS and DNA binding domain are described.

2. Figure 1- why not include activity of FOXO? e.g. which form of nuclear phospho-FOXO regulate gene expression? (or do all forms do this?).

Thank you very much for the comment on this important point. Phosphorylation by AMPK and MST1/JNK keeps FOXO in the nucleus. It has been shown that this leads to transactivation of the target genes. However, data does not directly show the phosphorylation state of FOXO when it is bound to the target gene and activates transcription. Therefore, I did not include the figure in which FOXO is bound to the promoter of the target gene. However, we find that the figure does not clearly show active/inactive Foxo1, therefore, we have revised the Figure 1 to clarify this point.

3. How good is the data with Resveratrol?- the life extending properties of this compound are frequently celled into question.

There are interesting papers showing Sirt1-Foxo1 signaling modulated by Resveratrol in various situation, including aging, insulin resistance, energy metabolism, and other diseases. However, we agree that more accumulated data may be required, especially to be included in a Review paper. Therefore, we have decided to exclude the resveratrol related story. 

4. Some formatting problems (paragraph width) around lines 409-411 and 455-463.

Thank you very much for pointing out the error. I have corrected the formatting problems.

5. C.elegans should be in italics (line 525).

Thank you very much for pointing out the error. I have revised C.elegans in italic.

5. The statements about "ecological integrity" in the conclusion are beyond the scope of this type of review.

I have rewritten the conclusion and toned down our view in FOXO function regarding “longevity” versus “ecological integrity”.